# Mind's Eye: Grounded Language Model Reasoning through Simulation

**Ruibo Liu**[1,2]**, Jason Wei**[1]**, Shixiang Shane Gu**[1]**, Te-Yen Wu**[2]**, Soroush Vosoughi**[2]
**Claire Cui**[1]**, Denny Zhou**[1]**, Andrew M. Dai**[1]
[1]Google Research, Brain Team, [2]Dartmouth College

## Abstract

Successful and effective communication between humans and AI relies on a shared experience of the world. By training solely on written text, current language models (LMs) miss the grounded experience of humans in the real-world—their failure to relate language to the physical world causes knowledge to be misrepresented and obvious mistakes in their reasoning. We present Mind's Eye, a paradigm to ground language model reasoning in the physical world. Given a physical reasoning question, we use a computational physics engine (DeepMind's MuJoCo) to simulate the possible outcomes, and then use the simulation results as part of the input, which enables language models to perform reasoning. Experiments on 39 tasks in a physics alignment benchmark demonstrate that Mind's Eye can improve reasoning ability by a large margin (27.9% zero-shot, and 46.0% few-shot absolute accuracy improvement on average). Smaller language models armed with Mind's Eye can obtain similar performance to models that are 100× larger. Finally, we confirm the robustness of Mind's Eye through ablation studies.

## 1 Introduction

> *"In questions of science, the authority of a thousand*
> *is not worth the humble reasoning of a single individual."*
>
> ——Galileo Galilei, 1632

*"Do objects fall proportionately to their weight?"* This famous question was once controversial until Galileo's Leaning Tower of Pisa experiment[1]—Galileo dropped two balls of different masses from the same height (i.e., *experiment*) and concluded that their time of descent was independent of their mass (i.e., *inductive reasoning*). Such an experiment-reasoning paradigm has been used by humans for centuries to ground reasoning on complicated problems (Newell, 1980) and transfer learned knowledge to unfamiliar domains (Novak & Gowin, 1984).

Current language models (LMs) follow a different path—by training on natural language, they attempt to reverse engineer the physical world, so that they are able to reason about it. Large-scale pre-trained LMs have achieved revolutionary performance on many tasks, such as solving math word problems (Roy & Roth, 2015; Ling et al., 2017; Cobbe et al., 2021) and commonsense reasoning (Talmor et al., 2022; Geva et al., 2021). However, these models do not experience firsthand the situations that are described by the language (McClelland et al., 2020), and lack the ability to find the correct answers by performing experiments like humans. As a consequence, when asked the same free fall question, one of the most widely-used LMs, GPT-3[2] (Brown et al., 2020)—though achieving superhuman performance in many reasoning tasks—will generate the wrong answer: *"The heavier object will fall faster."* (as shown in Figure 1). Due to the lack of grounded reasoning, current LMs also have issues in truthfulness (Lin et al., 2021) and factuality (Petroni et al., 2020).

---

[1]In *Physics*, Aristotle (384–322 BC) claims that the speed at which two identically shaped objects fall is directly proportional to their weights, which was later challenged by Aristotelian commentator John Philoponus.

[2]Specifically, we use text-davinci-002, which is the "most capable GPT-3 model" at the time of writing from OpenAI: https://beta.openai.com/docs/models/overview.

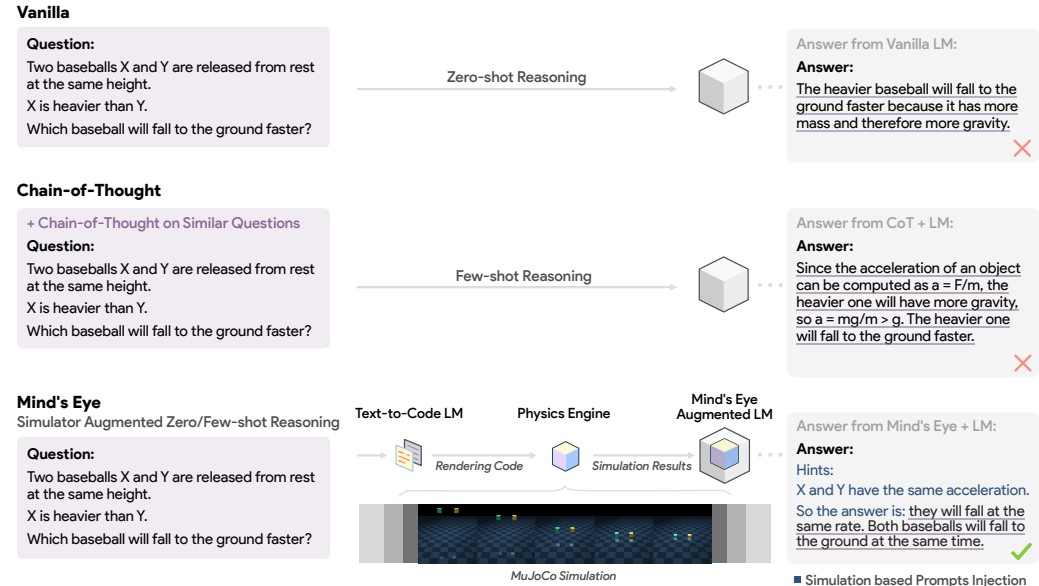

Figure 1: Current language models are still challenged by simple questions that require a good understanding of the physical world. The answer elicited by Chain-of-Thought can still be wrong if the required knowledge is missing or misrepresented in LMs. Mind's Eye, instead, enables grounded LM reasoning by directly simulating the scene in the given question. Then the LM can reason over the injected ground-truth rationale to generate the correct answers.

To tackle these problems, existing remedies include using improved prompting techniques, such as inserting hand-written decomposed reasoning steps in few-shot demonstrations (Wei et al., 2022; Zhou et al., 2022). These methods are inherently limited as their reasoning ability completely relies on the knowledge perpetuated in the LM—their performance could suffer if the knowledge learnt by the LM is incorrect (Petroni et al., 2019) or outdated (Dhingra et al., 2022). To incorporate external knowledge, retrieval-augmented LMs such as REALM (Guu et al., 2020), RAG (Lewis et al., 2020) and RETRO (Borgeaud et al., 2022), retrieve relevant documents as additional evidence for given questions, and may also fine-tune the LM on the question-document-answer triplets. However, the knowledge presented in written language is known to have *reporting bias* (Bisk et al., 2020), whereby some everyday unspoken facts or rarely seen (but practically possible) compositions are commonly missing in text (Paik et al., 2021).

Correct and complete understanding of properties and interactions in the physical world is not only essential to achieve human-level reasoning (Lake et al., 2017), but also fundamental to build a general-purpose embodied intelligence (Huang et al., 2022). In this work, we investigate to *what* extent current LMs understand the basic rules and principles of the physical world, and describe *how* to ground their reasoning with the aid of simulation. Our contributions are three-fold:

- We propose a new multi-task physics alignment dataset, UTOPIA, whose aim is to benchmark how well current LMs can understand and reason over some basic laws of physics (§2). The dataset contains 39 sub-tasks covering six common scenes that involve understanding basic principles of physics (e.g., conservation of momentum in elastic collisions), and all the ground-truth answers are automatically generated by a physics engine. We find that current large-scale LMs are still quite limited on many basic physics-related questions (24% accuracy of GPT-3 175B in zero-shot, and 38.2% in few-shot).

- We explore a paradigm that adds physics simulation to the LM reasoning pipeline (§3) to make the reasoning grounded within the physical world. Specifically, we first use a model to transform the given text-form question into rendering code, and then run the corresponding simulation on a physics engine (i.e., MuJoCo (Todorov et al., 2012)). Finally we append the

simulation results to the input prompts of LMs during inference. Our method can serve as a plug-and-play framework that works with any LM and requires neither handcrafted prompts nor costly fine-tuning.

- We systematically evaluate the performance of popular LMs in different sizes on UTOPIA before and after augmentation by Mind's Eye, and compare the augmented performance with many existing approaches (§4.2). We find Mind's Eye outperforms other methods by a large margin in both zero-shot and few-shot settings. More importantly, Mind's Eye is also effective for small LMs, and the performance with small LMs can be on par or even outperform that of 100× larger vanilla LMs.

## 2   UTOPIA BENCHMARKING

Humans are able to understand their physical environment and intuit rules about the world from embodied experience. The rules and principles behind the real world have been discovered as scientific laws—we humans have ingrained them as knowledge or intuition (Kaiser et al., 1986) to make reliable predictions on how observed events will unfold in day-to-day life (Kubricht et al., 2017). For example, when driving, we can anticipate when to brake when approaching a stop sign, using intuition or knowledge from Newton's second law of motion. We also know it would be a disaster to collide with a heavy truck, not only in terms of our knowledge on the conservation of momentum (i.e., the lighter object will have a greater velocity after collision), but also from our embodied experience of collision in everyday life.

We are thus inspired to design a physics alignment dataset that covers this knowledge, aiming to benchmark to what extent current LMs understand basic physical concepts and rules. As shown in Table 1, we choose six representative scenes, mainly from textbooks (e.g., high-school Physics). The sub-tasks are defined based on the composition of observed and queried concepts. For example, one task in a motion scene could be: given the observed *acceleration* of the two objects with the same mass, please answer what is the relationship of *forces* applied on them. In total we have 39 sub-tasks across different scenes, and each sub-task contains various hand-designed questions whose language style is similar to that of textbooks as well.

Table 1: We propose UTOPIA, a multi-task physics alignment dataset, investigating the grounded reasoning ability of LMs on 39 sub-tasks. Unlike many other datasets, UTOPIA deliberately describes the questions in relative relations (e.g., *greater than*) instead of absolute numbers (e.g., 3.5 m/s), to approximate human's perceptual sensing ability in real world. The ground-truth answers to the questions are generated by the physics engine, which makes it easy to scale UTOPIA to larger sizes.

| Scenes | (Simplified) Sample Questions | Concepts | # Tasks |
|---|---|---|---|
| Motion | Amy pulls two sleds X and Y with the same force. X has a greater mass than Y. Friction can be ignored. Which one has a greater acceleration after the same period of time? | mass force velocity | 6 |
| Friction | Two boxes X and Y move at the same velocity. We only consider kinetic frictions, and X undergoes a smaller friction than Y. Which one has a greater velocity after the same period of time (before stop)? | mass velocity friction | 6 |
| Free fall | Two balls are dropped from the same height. Y has a greater mass than X. We ignore the air resistance. Which one will hit the ground earlier? | mass height energy | 6 |
| Projection | Jason throws two baseballs X and Y at the same height horizontally. They have the same mass, but X has a greater initial horizontal velocity. Which one will hit the ground earlier? | velocity mass energy | 6 |
| Collision | Two marbles X and Y of the same mass move towards each other. X and Y have the same magnitude of velocity, and the collision is elastic. Which one will have a greater velocity after collision? | velocity mass momentum | 6 |
| Incline | Two blocks of metal X and Y are released from a certain height on a slick slope. Y has a greater mass than X, and the friction can be ignored. Which one will have a greater velocity after the same period of time? | mass height friction | 9 |

Table 1 exemplifies some samples in UTOPIA. We deliberately choose to use relative comparison (e.g., *"greater than"*, *"smaller than"*, *"the same as"*; text in purple) rather than actual numbers to

describe the physical properties, since we are thus able to disentangle the effects from numeracy (i.e., the gain on reasoning is not attributed to better memorization on numbers, which has been reported as "shortcuts" used by LMs (Patel et al., 2021)). This setting is also different from those in mathematical reasoning tasks (e.g., GSM8k (Cobbe et al., 2021)), where the decomposed reasoning path is typically the procedure of plugging different values into equations—the LM might be able to solve these problems by symbolic manipulation (Razeghi et al., 2022) rather than actual reasoning.

Most existing physics alignment datasets use vision as the primary modality, such as images (Zellers et al., 2019), animations (Wu et al., 2017), or videos (Piloto et al., 2022), which loses the flexibility to run on LMs which only takes text input. PIQA (Bisk et al., 2020) and MMLU-Physics (Hendrycks et al., 2021) are popular physics reasoning datasets used for LM benchmarking; however, their sizes are naturally limited because of required human annotations (e.g., only 206 samples are on physics in MMLU, with college and high school level questions combined). UTOPIA differs from all these datasets as it leverages a physics engine to generate data—in theory we can obtain *unlimited* samples—and each sample has reliable ground-truth supported by actual simulation. Although in the present work we only take the text-form data for LM benchmarking, the corresponding simulation videos during data generation have been recorded as data for future multi-modality research.

## 3 MIND'S EYE

As shown in Figure 1, Mind's Eye comprises three main components, a text-to-code LM as the front-end, a physics simulation engine (i.e., MuJoCo) as the back-end, and a foundation model (Bommasani et al., 2021) for general reasoning. We detail the implementation of Mind's Eye as below:

**Text-to-Code Converter.** The objects and dynamics of the simulation is manifested by the rendering code fed into MuJoCo. The rendering code is written in a type of XML file named MCJF[3], where the physics properties can be easily controlled by changing some key-value pairs. For example, to change the mass of an object to 10, the line of rendering code needed is `geom.set('mass', '10')`. We use actual values to express the relative relationships in UTOPIA (e.g., *"greater"* will be translated to 10 and 1 for the values of the properties to be set). We create rendering templates for each sub-task of UTOPIA, and use programs to generate a dataset with 200,000 text-code pairs. In each pair, the question in text is appended to the top of the XML code as comments. We then train decoder-only LMs from scratch to learn how to generate the rendering code given the question in comments auto-regressively. We leverage the BPE vocabulary set from GPT-2 (Radford et al., 2019) and extend it by several special tokens to represent repeating tabs or spaces. Besides fine-tuning on the dataset with text-code pairs, we also pre-train the model on the C4 dataset (Raffel et al., 2019a) to enhance the model's understanding on natural language. All the training is on TPU-v3 Pods and the resulting models have 0.3B and 1.5B parameters (used as default). See §4.1 for training details.

**Simulation Augmented Prompting.** Once receiving the rendering code, the physics engine will run the corresponding simulation to get the ground-truth outcome. The program that triggers the simulation will also parse the outcome into text-form prompt injections (e.g., *"Hints: Two baseballs take the same time to hit the ground."*, as shown in Figure 1). The injection combined with the question will be fed to the foundation model, with which LMs can ground their reasoning with the physical world rendered by the physics engine. We present more details of this procedure in §A.1.

The intuition behind Mind's Eye is to imitate the experiment-reasoning paradigm; however, we leverage quick and cheap physics simulation as an alternative to actual experiments in physical world. The cognitive analog for Mind's Eye might be the mental visualization process, also known as "the mind's eye" (Battaglia et al., 2013; Hegarty, 2004), which often relates to motor processes (Wexler et al., 1998) during embodied reasoning (Nathan et al., 2021).

**Discussion: Why does Mind's Eye work?** Table 2 shows the comparison between Mind's Eye and two other methods in the formulation of the grounding process during LM inference. Assuming knowledge of the physical world aligns with the distribution $p_{\text{World}}$, the Zero-shot Reasoner (Kojima et al., 2022) which uses *"Let's think step by step."* in prompts can be extended to any number of new tasks. However, its reasoning ability will be compromised if the knowledge in LMs is incorrect or outdated. Similarly, incorporating handcrafted reasoning steps rather than a generic phrase, Chain-

---

[3]Docs for MJCF: `https://mujoco.readthedocs.io/en/latest/XMLreference.html`

Table 2: Comparison in formulation of how the LM inference process is grounded (given question $x$ generating answer $y$). With the aid of a simulator (i.e., MuJoCo), Mind's Eye is not only scalable (not requiring human annotation) but also well-grounded with the physical world.

| Method Name | Formulation | Scalable? | Grounded? |
|---|---|---|---|
| Zero-shot Reasoner | $y \leftarrow \arg\max_{\hat{y}} \text{LM}(\hat{y}|x, \text{``Let's think step by step''})$ | ✓ | ✗ |
| Chain-of-Thought | $y \leftarrow \arg\max_{\hat{y}} \text{LM}(\hat{y}|x, \text{Chain-of-Thought} \sim p_{\text{Human}})$ | ✗ | ? |
| **Ours:** Mind's Eye | $y \leftarrow \arg\max_{\hat{y}} \text{LM}(\hat{y}|x, \text{Simulator}(x, \hat{y}) \sim p_{\text{World}})$ | ✓ | ✓ |

of-Thought (Wei et al., 2022) is able to elicit LM reasoning in a more explicit way; however, its performance is reported to be sensitive to the quality of human annotated reasoning steps (Lampinen et al., 2022; Dasgupta et al., 2022) (i.e., $p_{\text{Human}}$ is not similar to $p_{\text{World}}$). The dependence on human annotation also limits its scalability.

Mind's Eye overcomes these issues by including a simulator into the reasoning pipeline. Given the question, the simulator (i.e., the physics engine) returns the most likely outcome based on its encoded world knowledge. Since the simulator is accurate enough to approximate the physical world, the prompt injection of Mind's Eye basically serves as a scoring machine, which puts probability mass on the answer that is best aligned with the rules of physics—the LM reasoning over the injected rationales is thus grounded. Mind's Eye is also scalable since the whole pipeline is automated.

Besides scalability and grounded reasoning, Mind's Eye is also efficient, since it delegates domain-specific knowledge to external expert models (i.e., the MuJoCo engine for expert physics knowledge) (Du et al., 2022; Shazeer et al., 2017), which decouples general reasoning from domain specific knowledge. The size of the LM can thus be significantly shrunk since it removes the burden of memorizing all the domain-specific knowledge. Experiments find that 100× smaller LMs augmented with Mind's Eye can achieve similar reasoning capabilities as vanilla large models, and its prompting-based nature avoids the instability issues of training mixture-of-expert models (Zoph et al., 2022). The compatibility with small LMs not only enables faster LM inference, but also saves time during model saving, storing, and sharing.

## 4 EXPERIMENTS

### 4.1 EXPERIMENTS SETTINGS

**Data and Model.** For the convenience of benchmarking on huge LMs, we prepare 100 samples for each sub-task, resulting in a dataset with about 3,900 samples. We use this version of UTOPIA for evaluation across the paper. The MuJoCo simulations can achieve 171 fps on one A6000 GPU, and generating 100 simulations of a 2 seconds collision scene takes 0.67s. For LMs, besides GPT-3, we have also tested Pathway Language Model (PaLM) (Chowdhery et al., 2022) on 8B, 62B, and 540B checkpoints. All experiments for PaLM are run on TPU-v4 Pods.

**Training Details.** Training of the JAX-based text-to-code LMs runs on TPU-v3 Pods. The learning rates we use for training 0.3B and 1.5B LMs on C4 are {3.0e-4, 1.8e-4}, which are switched to {1.8e-4, 0.5e-4} when fine-tuning on the text-code pairs. We use cosine annealing to control learning rate over time with fixed warm-up steps (3k). As mentioned in §A.2, we have also fine-tuned 62B PaLM to study task generalization, which takes about 25 minutes on 64 TPU-v4 chips for each task.

### 4.2 BENCHMARK RESULTS ON UTOPIA

As shown in Figure 4.2, we run experiments on UTOPIA with GPT-3 and PaLM of different sizes, ranging from 340M (GPT-3 Ada) to 540B (PaLM). Although larger LMs perform consistently better than smaller LMs in both zero-shot and few-shot settings ($n = 5$), reasoning ability seems to plateau after a certain size, especially in the few-shot setting. In other words, the scaling curve of vanilla few-shot is nearly flat. One interpretation could be that few-shot demonstrations have managed to trigger effective in-context learning (e.g., for learning the answer format), but the lack of grounded reasoning becomes the bottleneck for further improvement.

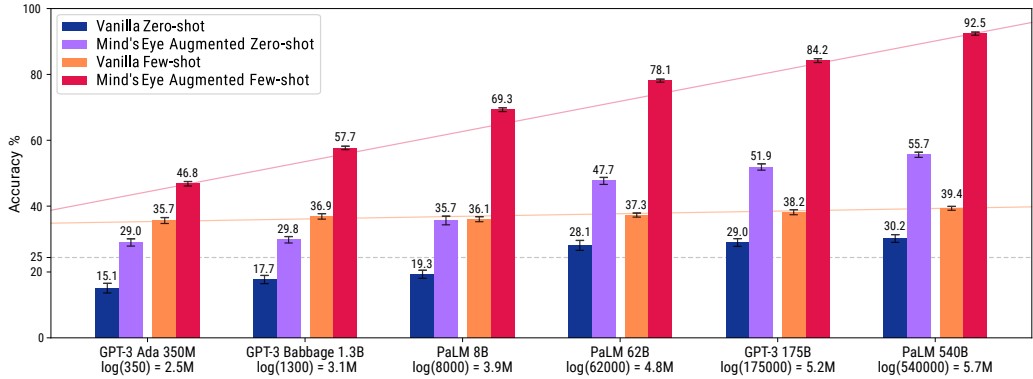

Figure 2: Scaling law of reasoning on UTOPIA when benchmarking LMs in different sizes (log scale). Smaller LMs augmented with Mind's Eye can achieve on par or even outperform larger vanilla LMs in both zero-shot and few-shot settings (e.g., 29.8% for GPT3 1.3B + Mind's Eye vs. 29% for vanilla GPT-3 175B in zero-shot). In few-shot settings ($n = 5$), the scaling potential is unlocked when LMs are augmented with Mind's Eye (red line), as the scaling curve is nearly flat in vanilla few-shot mode (orange line), which demonstrates the power of incorporating knowledge from simulations. On average across all sizes, the gain from Mind's Eye is greater in large LMs than in small ones, and greater in few-shot (34.2%, absolute) than in zero-shot settings (18.4%, absolute).

Mind's Eye, however, unlocks the ability to scale (red line in Figure 4.2) by adding knowledge from physics engine simulations. Since the correctness of the simulation results is guaranteed by the physics engine, the reasoning of the large foundation model is thus well grounded. Compared with solely using knowledge perpetuated in LMs (results with vanilla LMs), Mind's Eye is able to boost the reasoning ability of LMs by 18.4% in zero-shot and 34.2% in few-shot settings. Interestingly, smaller LMs augmented with Mind's Eye can achieve similar or even better performance than vanilla larger LMs. For example, the accuracy of GPT-3 Babbage 1.3B with Mind's Eye in zero-shot is 29.8%, while that of vanilla GPT-3 Davinci 175B in zero-shot is 29%. This finding demonstrates the effectiveness of the Mind's Eye paradigm decoupling *experimenting* from *reasoning*—where the domain specific tool is responsible for providing ground-truth results, and the LM can mainly focus on general reasoning, whose size can thus be largely decreased.

**Comparison with other reasoning enhanced techniques.** In Table 3, we compare Mind's Eye with other methods that improve the reasoning abilities of LMs (i.e., GPT-3 175B). In addition to a) **Chain-of-Thought** (Wei et al., 2022), we consider other prompt-based methods such as b) **Zero-shot Reasoner** (Kojima et al., 2022), which uses *"Let's think step by step."* in prompts to induce the decomposed reasoning steps of LMs; c) **Self-Consistency Decoding** (Wang et al., 2022), which is an ensemble technique that decodes multiple reasoning paths concurrently to improve the performance of Chain-of-Thought; the recent study d) **DiVerSe** (Li et al., 2022b) achieves new SotAs on many reasoning tasks, by using pre-trained verifiers to weight good answers more than bad ones in each step decoding. Besides RAG (Lewis et al., 2020) that retrieves knowledge from the memory of 75GB documents, we also consider other fine-tuned LMs which attempt to optimize reasoning ability from different perspectives, such as e) task-scaling **T0 (version pp)** (Sanh et al., 2021), which fine-tunes T5 (Raffel et al., 2020) on thousands of prompts to make LMs better understand input prompts, especially in zero-shot settings, and f) model-scaling **Minerva** (Lewkowycz et al., 2022), which fine-tunes PaLM (Chowdhery et al., 2022) 540B on a newly collected dataset that contains scientific and mathematical documents (e.g., arXiv papers) to improve quantitative reasoning. Though most experiments are running on GPT-3 175B, we also explore the scaling effect by using the 1.3B Babbage model. The 'grounding gain' is the absolute accuracy difference for the 175B model augmented with and without Mind's Eye. Unless otherwise stated, we use the default parameter settings recommended by competitor methods.

Results shows that Mind's Eye outperforms other methods in both zero-shot and few-shot settings significantly, even if using a relatively smaller LM (i.e., GPT-3 Babbage 1.3B). Comparing results on

Table 3: Comparison of Mind's Eye and other methods on UTOPIA benchmarking. Zero-shot Reasoner (Kojima et al., 2022), Chain-of-Thought (Wei et al., 2022) are two prompt-based methods that can elicit reasoning in large-scale LMs. Self-consistency (Wang et al., 2022) and DiVerSe (Li et al., 2022b) are decoding-time optimization techniques for LMs reasoning. RAG (Lewis et al., 2020) is a retrieval augmented LM, while T0 (Sanh et al., 2021) and Minerva (Lewkowycz et al., 2022) are fine-tuned LMs to improve reasoning ability by task-scaling and model-scaling. We present results of Mind's Eye on GPT-3 175B/1.3B, and find that a 100× smaller LM can outperform a vanilla 175B model (*ref.*) when armed with Mind's Eye. Interestingly, fine-tuning on prompts to better follow human instructions, Instruct-GPT (Ouyang et al., 2022) can achieve nearly perfect physics alignment in few-shot. We also annotate the grounding gain of Mind's Eye against vanilla GPT-3 175B.

| | Motion | | Friction | | Free Fall | | Projection | | Collision | | Incline | | Average | |
|---|---|---|---|---|---|---|---|---|---|---|---|---|---|---|
| Existing Methods | Zero | Few | Zero | Few | Zero | Few | Zero | Few | Zero | Few | Zero | Few | Zero | Few |
| GPT-3 175B (*ref.*) | 21.5 | 34.3 | 24.3 | 40.0 | 22.2 | 44.5 | 27.0 | 31.8 | 26.5 | 44.8 | 22.6 | 33.9 | 24.0 | 38.2 |
| GPT-3 1.3B | 15.8 | 34.3 | 16.7 | 33.2 | 6.7 | 51.8 | 18.5 | 33.0 | 13.5 | 33.5 | 19.6 | 32.4 | 15.1 | 36.4 |
| Instruct-GPT 175B | 76.0 | 85.3 | 60.2 | 83.2 | 39.6 | 47.6 | 39.8 | 50.0 | 49.0 | 76.8 | 37.1 | 68.6 | 50.3 | 68.6 |
| **Better Prompting** | | | | | | | | | | | | | | |
| Zero-shot Reasoner | 25.0 | - | 35.0 | - | 31.6 | - | 21.6 | - | 32.0 | - | 28.5 | - | 29.0 | - |
| Chain-of-Thought | - | 41.5 | - | 42.2 | - | 53.0 | - | 40.3 | - | 38.0 | - | 41.6 | - | 42.8 |
| **Optimized Decoding** | | | | | | | | | | | | | | |
| Self-Consistency | - | 38.9 | - | 44.5 | - | 57.2 | - | 48.2 | - | 48.8 | - | 43.9 | - | 46.9 |
| DiVerSe | 20.2 | 33.1 | 17.3 | 29.6 | 24.3 | 37.6 | 23.1 | 35.6 | 25.4 | 39.3 | 24.5 | 37.8 | 22.4 | 35.5 |
| **Augmented LMs** | | | | | | | | | | | | | | |
| RAG (0.4B + 75GB) | 7.4 | - | 8.3 | - | 10.2 | - | 5.2 | - | 11.4 | - | 8.5 | - | 8.5 | - |
| T0-pp (11B) | 12.3 | 29.4 | 14.3 | 34.1 | 11.4 | 25.5 | 13.7 | 27.1 | 9.5 | 16.8 | 17.2 | 30.9 | 13.1 | 27.3 |
| Minerva (540B) | 24.5 | 45.3 | 29.8 | 41.0 | 6.2 | 15.5 | 36.3 | 51.0 | 23.3 | 46.3 | 24.3 | 43.4 | 24.1 | 40.4 |
| **Ours:** Mind's Eye | | | | | | | | | | | | | | |
| w/. GPT-3 175B (★) | 52.0 | 82.5 | 57.0 | 82.1 | 30.3 | 89.8 | 65.7 | 83.8 | 54.0 | 79.0 | 52.3 | 87.8 | 51.9 | 84.2 |
| w/. GPT-3 1.3B | 33.7 | 41.7 | 35.0 | 42.3 | 12.8 | 70.0 | 34.8 | 40.0 | 30.5 | 45.0 | 31.8 | 41.3 | 29.8 | 46.7 |
| w/. Instruct-GPT 175B | 100.0 | 100.0 | 97.8 | 100.0 | 54.9 | 94.8 | 91.8 | 99.8 | 61.8 | 99.8 | 99.4 | 100.0 | 84.3 | 99.1 |
| Grounding Gain (%) | 30.5 | 48.2 | 32.7 | 42.1 | 8.1 | 45.3 | 38.7 | 52.0 | 27.5 | 34.2 | 29.7 | 53.9 | 27.9 | 46.0 |

GPT-3 175B and 1.3B, we can conclude that 1) larger LMs can better leverage Mind's Eye, especially in few-shot settings (46.0% vs. 10.3% average grounding gain), probably because larger LMs are more capable of general reasoning, and 2) solely scaling-up models is not adequate for reliable grounded reasoning performance. For example, when using a 100× larger LM (1.3B → 175B), the few-shot accuracy is merely boosted by 1.8% (absolute; 36.4% → 38.2%) if using vanilla LMs, but that can be boosted by 37.5% (absolute; 46.7% → 84.2%) if the LMs are augmented by Mind's Eye.

Note that Instruct-GPT(Ouyang et al., 2022) augmented with Mind's Eye is able to achieve nearly perfect performance in few-shot settings (68.6% → 99.1%). This result is promising because it demonstrates the ideal alignment is achievable if the LM is given proper reasoning rationale and has good understanding of the questions (as Instruct-GPT is optimized for instruction following). The improvement from simply better prompting or decoding methods is limited, since their reasoning completely relies on the internal knowledge perpetuated in the LMs, while the knowledge induced from the LMs could be factually wrong. Among augmented LMs, 540B Minerva is the best performing one but still falls behind Mind's Eye + 175B GPT-3, mainly due to the lack of grounded experience for reasoning. We also find the retrieved evidence of RAG sometimes cannot answer the question, even though it includes entities mentioned in the question, and RAG cannot function well in few-shot settings since the retriever module (i.e., DPR (Karpukhin et al., 2020)) was not pre-trained to handle context that contains multiple question-answer pairs. We have also discussed the domain generalization ability of Mind's Eye in §A.2, and error analysis in §A.3.

## 4.3 ABLATION STUDY

**Do we really need simulation?** In Table 4, we show the performance if 1) we randomly alter the simulation results in the prompts to mismatch the physics property asked (e.g., asking velocity but including acceleration results in prompts), and 2) we delete the trigger words at the beginning of the prompts (i.e., *"Hints:"*). We also test what would happen if we ground the reasoning on incorrect

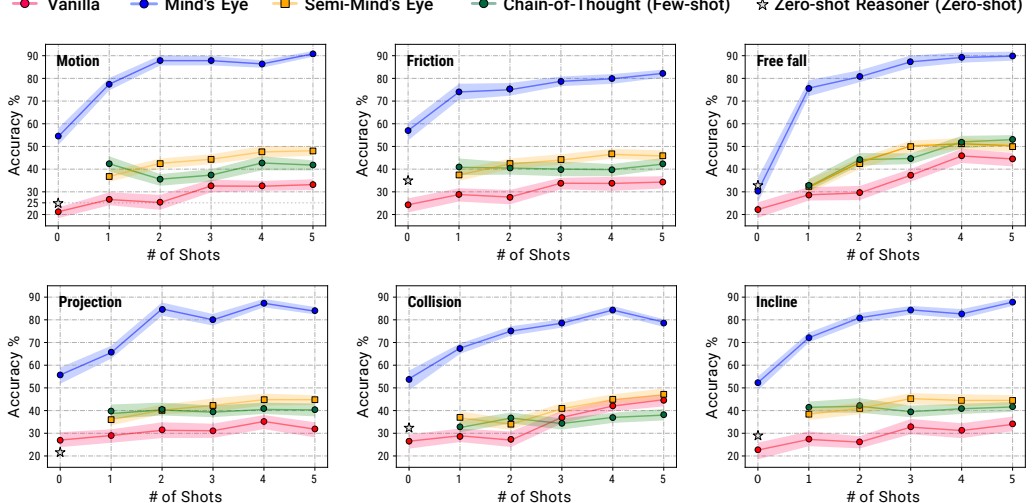

Figure 3: The dynamics of reasoning ability on UTOPIA when we increase the number of shots from zero to five, to examine whether we achieve similar performance with more in-context demonstrations instead of using external simulators (as Mind's Eye).

simulation results (e.g., Simulation indicates *greater* but we use *smaller* in prompts). We find reasoning on mismatched simulation results causes substantial performance drops, while including wrong simulation results will further deteriorate the performance, probably because in this case the reasoning is misguided even if the vanilla LM can answer the question correctly. We take these results as evidence that the correct simulation results are crucial for grounded LM reasoning. Missing trigger words have marginal influence on the performance, which confirms again that reasoning performance mainly depends on the existence and correctness of the simulation results.

Table 4: Ablation study on the simulation results and the trigger words of Mind's Eye to understand their effects. We also study whether the correctness of simulation will affect the reasoning performance.

| Ablation Settings | Zero-shot | Few-shot |
|---|---|---|
| Mind's Eye (*default*) | $51.9_{1.73}$ | $84.2_{0.79}$ |
| Mismatched simulation results | $30.4_{1.65}$ | $54.3_{1.28}$ |
| Missing trigger words | $51.0_{1.12}$ | $83.0_{0.73}$ |
| Incorrect simulation | $24.6_{1.73}$ | $39.6_{2.89}$ |

Table 5: The effect of using different sizes of text-to-code models (T2C) with GPT-3 175B/1.3B as the foundation model (FM) in the zero-shot and few-shot settings.

| LM Size (T2C + FM) | Zero-shot | Few-shot |
|---|---|---|
| 0.3B + 1.3B | $25.1_{1.77}$ | $43.3_{1.12}$ |
| 0.3B + 175B | $48.3_{1.61}$ | $82.1_{0.89}$ |
| 1.5B + 1.3B | $29.8_{1.65}$ | $46.7_{0.82}$ |
| 1.5B + 175B (*default*) | $51.9_{1.53}$ | $84.2_{0.79}$ |

**Can few-shot replace simulation?** Given enough in-context demonstrations, can the LM learn how to make its reasoning grounded *internally* without relying on *external* simulations? In Figure 3, we present the dynamics of reasoning performance by gradually including more and more in-context demonstrations in few-shot settings on both vanilla LMs and Mind's Eye augmented LMs. We also design a semi-augmented baseline, Semi-Mind's Eye, which has the Mind's Eye style in-context demonstration, but the final shot omits the simulation result—in other words, the LM has to perform reasoning on the generated grounding rationale by itself in the final step. This baseline differs from vanilla few-shot as it incorporates some simulation results from the physics engine, and it differs from Chain-of-Thought (CoT) since the reasoning steps of CoT are written by humans.

The result demonstrates neither vanilla few-shot (red line) nor semi-augmented few-shot (yellow line) can provide as good an improvement as Mind's Eye (blue line). We also find the few-shot reasoning performance of CoT (green line) and Semi-Mind's Eye (yellow line) has some instabilities which depends on whether the few-shot demonstrations happen to have similar reasoning steps as the given question (final step). The effectiveness of Mind's Eye seems to echo the findings in Zhao et al. (2021),

which confirms that the LM tends to ground its reasoning on "recent context", which is coincidentally the simulation results of the given question.

**Using a smaller text-to-code LM?** Text-to-code LMs convert text-form questions into rendering code. To study its scaling effect, we explore several combinations of using text-to-code LMs and pretrained LMs in different sizes. As shown in Table 5, we find using a smaller text-to-code LM (0.3B) has slightly smaller impact for larger pretrained LMs, as the accuracy decreases by 4.7% for 1.3B pretrained LM but 3.6% for 175B in zero-shot. We also find the conversion error is somehow mitigated by few-shot demonstrations (e.g., the accuracy decreases by 3.6% in zero-shot but by 2.1% in few-shot when the 175B model uses a smaller T2C LM), indicating the benefits on robustness when using a larger pretrained LM.

## 5    RELATED WORK

**Grounded Reasoning.** Early attempts to relate language understanding (Winograd, 1972) or learning (Roy & Pentland, 2002) to the physical world mostly rely on manually created linguistic and physical rules (Hermann et al., 2017; Berant et al., 2013). Recent studies have claimed that pre-trained large-scale LMs have already memorized enough world knowledge (Roberts et al., 2020; Brown et al., 2020), and enhanced reasoning ability can be achieved by proper prompting (Nye et al., 2021; Wei et al., 2021; Sanh et al., 2021). Besides adding decomposed reasoning steps (Wei et al., 2022; Zhou et al., 2022), previous work has tried to add hand-written task descriptions (Raffel et al., 2019b) and targeting formats (Marasović et al., 2021) to the prompts, but such human annotation is often costly and expensive at scale (Li & Liang, 2021; Shin et al., 2020). Mind's Eye, instead, presents a new paradigm to ground LM reasoning, via automated simulation rather than human crafted rationales.

**Augmented Language Models.** Inspired by the evidence that humans require sensory perception for grounded reasoning (Sachs et al., 1981), previous work has tried to augment text inputs to LMs with audio signals (Tsai et al., 2019) or visual perception (Bakhtin et al., 2019; Yi et al., 2018), for improved game playing (Lee et al., 2022), faster language learning (Hill et al., 2020), or better decision making in general (Reed et al., 2022). Our approaches seemingly echoes these findings as we leverage the simulation results as the extra sensory input. To endow LMs with updated knowledge, TALM (Parisi et al., 2022) fine-tunes LMs interactively with augmented inputs from API calls, which is rather costly compared to prompt-based Mind's Eye. PaLM-SayCan (Ahn et al., 2022) uses the pre-trained PaLM 540B to help robots better understand complex instructions that require reasoning—Mind's Eye can be viewed as the reverse: it infuses knowledge from external tools (i.e., the MuJoCo physics engine) into LMs, to further unlock the the reasoning capabilities of LMs.

**Modeling the World.** Learning by trial and error, humans seem able to learn enormous amounts of common sense about how the physical world works (Lerer et al., 2016)—such observation has inspired the idea of developing a neural model to model the world (LeCun, 2022; Craik, 1967). World models should be capable of planning (Li et al., 2022a), predicting (Amos et al., 2018), and reasoning (Ullman et al., 2017) through interactions with the world. Similarly, Mind's Eye proposes a paradigm that *reasons* over the experimental results *predicted* by simulation, and the experiments are *planned* beforehand by the text-to-code LM. Using simulated environments to help learning has been widely adopted by research in robotics (MineRL; Guss et al. (2019)) or computer vision (Kubric; Greff et al. (2022)), while our focus is grounded reasoning in the form of natural language.

## 6    CONCLUSION AND FUTURE WORK

We have proposed Mind's Eye, a novel paradigm that enables LMs to ground their reasoning with simulations of the world. We conclude that Mind's Eye is not only effective and scalable but also efficient, as it is able to boost the reasoning performance of small-scale LMs significantly, requiring neither handcrafted prompts nor costly fine-tuning.

We believe the idea of including simulation into the reasoning pipeline can be easily extended to other domains or applications, especially where simulation engines already exist. For example, instead of the physical world, one can use a simulation of economic changes or thermodynamics, aiding policy making or engineering. The dynamic nature of Mind's Eye where we generate grounding evidence unlocks the scaling potential of these models.

## ETHICS AND REPRODUCIBILITY STATEMENT

The goal of Mind's Eye is to present a general-purpose paradigm that incorporates knowledge from simulation by professional tools into the LM reasoning pipeline. Though we have already seen significant improvement in reasoning with Mind's Eye grounding, the performance can be affected by how accurate the simulation is. Unqualified simulators may result in wrong simulation results, grounding with which will lead to even worse results than reasoning without grounding. The artifacts of the LM itself can also affect the final performance since it is responsible for the general reasoning (Bai et al., 2022; Liu et al., 2021; 2022). Furthermore, our experiments and analysis are done in English, and therefore we do not claim that our reasoning paradigm will generalize across all languages, although our framework has the potential to be extended to other languages by using multilingual LMs. For reproducibility, we run experiments mainly with publicly available LMs (e.g., GPT-3) and choose baseline methods that have open-sourced implementation.

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

# A    APPENDIX

## A.1    DETAILS ABOUT MIND'S EYE

In Figure A.1 we have presented the whole procedure on how a text question is converted into rendering code for simulation, and how the simulation results are parsed by the manager program to produce simulation based prompt injection.

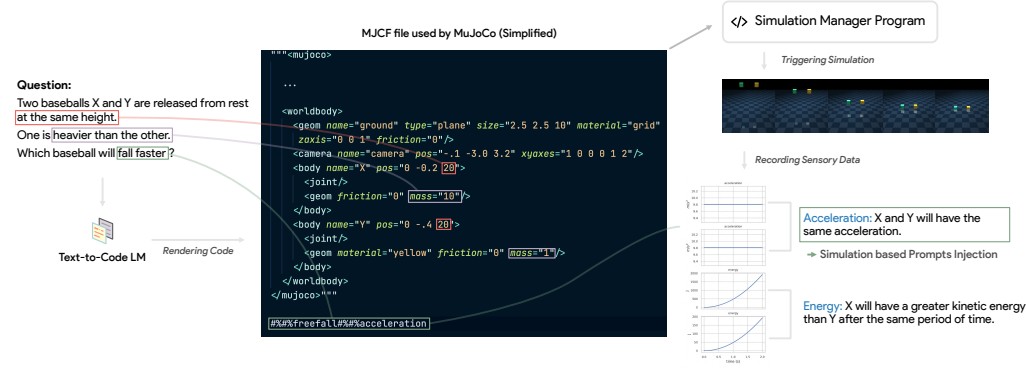

Figure A1: We demonstrate the whole pipeline of how Mind's Eye generates the simulation based prompts injection for a given physics question. We highlight those attributes in the rendering code that reflect the relative relationship in a UTOPIA question. A simulation manager program is responsible for triggering simulation and parse the results. Here we only show two of physics properties that the manager records for demo purpose.

Trained on many text-code pairs, the text-to-code LM is able to generate the corresponding rendering code for the given physics related question. Note that in the generated rendering code, besides the MJCF file itself, in the last line there are special signs #%#% to record the scene name and the physics attributes of interest—these meta-information are added during fine-tuning data (text-code pairs) synthesis stage prepared for the text-to-code LM. After fine-tuning, the text-to-code LM should be able to generate such information.

Once the simulation manager program receives the rendering code, it will execute the simulation based on it. Sensory data such as velocity, acceleration, energy, etc. will be recorded (via MuJoCo APIs). Finally, the manager program will parse the data in terms of the physical properties of interests (coming with the rendering code, marked by #%#%). For example, in Figure A.1, the sensory data is recorded along with the free fall simulation, and since the asked property is "acceleration", the manager program will extract the sensory ground-truth data only on acceleration, and draws the conclusion *"X and Y will have the same acceleration."* by filling in pre-defined "conclusion templates" (e.g., *"The {physics property} of X will be {greater/smaller} than that of Y."*) with observations. Some trigger words will be concatenated before (e.g., *"Hints:"*) the generated conclusion. The connection words (e.g., *"So the answer is:"*) are also added to elicit the final answer. The whole concatenation will be appended after *"Answer:"* as grounded rationale for LM reasoning (as shown in Figure 1 in the main body of the paper).

## A.2    SCENE GENERALIZATION WITH MIND'S EYE

By injecting simulation based prompts during LM inference, Mind's Eye does not need fine-tune the LM every time it encounters new contexts. This feature is appealing as we expect Mind's Eye can serve as a powerful tool for building autonomous intelligence that can ground its reasoning or decision making even in unfamiliar contexts. To clearly show the advantages of Mind's Eye on scene generalization, we compare the performance of fine-tuned PaLM 62B with that of vanilla PaLM 62B armed with Mind's Eye.

As shown in Figure A.2, we find the fine-tuned model can achieve good performance on in-domain data, but its performance is quite limited in scenes it has not been fine-tuned on. For example, in Figure A.2 (a), fine-tuned on the Motion scene of UTOPIA can obtain 70% in domain alignment accuracy, but the accuracy decreases to 24% if we test this model on the Projection scene (> 50% performance drop). However, as shown in Figure A.2 (b), Mind's Eye enables the LM to perform consistently well, even if the LM has never seen the scene before.

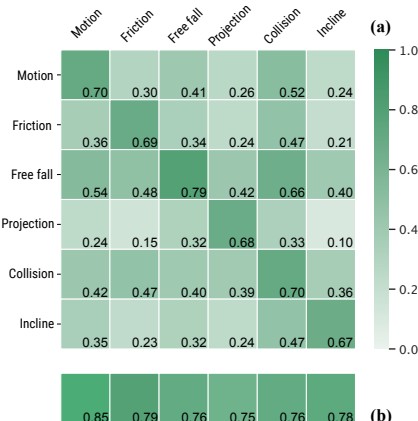

Figure A2: We compare the performance of (a) fine-tuned 62B PaLM on six scenes of UTOPIA data respectively with (b) vanilla 62B PaLM armed with Mind's Eye in zero-shot. Though fine-tuning can achieve good performance on in-domain data, it can hardly generalize to OOD cases (i.e., scenes on which the LM has not been fine-tuned). Prompting based Mind's Eye can enable large LMs generalize to unseen new scene with consistent alignment performance even in zero-shot settings. In (a), the performance of fine-tuned PaLM 62B on in-domain test sets is in diagonal, while that of out-of-domain is the rest. The zero-shot performance of PaLM 62B + Mind's Eye in six scenes is listed in (b).

## A.3   MIND'S EYE ERROR ANALYSIS

In Figure A.3 we show three error cases when using Mind's Eye. Ignorance error refers to the reasoning apparently ignores the given grounded rationale. Recency bias means the LM tends to extract the final answer from nearby words (e.g., the object Y). "I dont́ know" error are the cases where the LM simply generates non-sense when unable to reason about the question. All these cases are much less common when we run experiments on larger LMs.

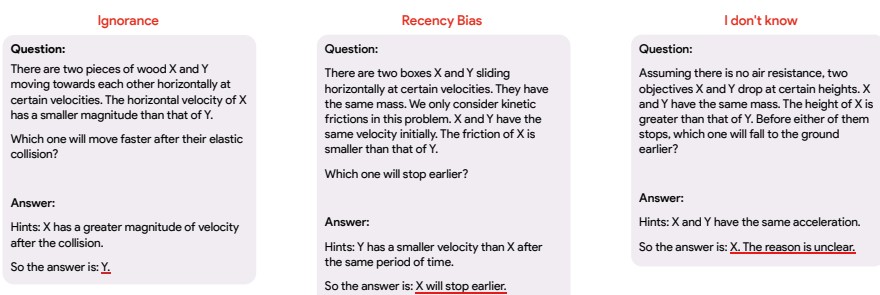

Figure A3: We exemplify several error cases when using Mind's Eye. In general, we find these errors mostly happen in small LMs, because their reasoning ability over given evidence is limited.

