# OpenReview forum: "Mind's Eye: Grounded Language Model Reasoning through Simulation"
_ICLR.cc/2023/Conference — ICLR 2023 poster_

### Official Review · Reviewer_46te · 2022-10-23

**Confidence:** 3
**Correctness:** 4
**Technical Novelty And Significance:** 2
**Empirical Novelty And Significance:** 3
**Recommendation:** 6

**Clarity, Quality, Novelty And Reproducibility:**

The paper is clearly written and of high quality. As mentioned in the previous section, the proposed method is conceptually a combination of two previous lines of work. Nonetheless, this combination is novel and reveals unique insights about using LM in a domain constrained by a lack of knowledge.


**Strength And Weaknesses:**

Strength:
- The proposed method is simple and effective at improving the zero-shot and few-shot physics reasoning performances of LMs.
- Thorough experiments at various scales are performed.
- The proposed dataset can be generated automatically from simulation and potentially extensible to more complicated scenarios.

Weaknesses:
- The current form of Mind's Eye seems very specifically designed for the proposed dataset UTOPIA, which only contains simple physics questions that compare 2 objects and are generated by physics simulations to begin with. The generalization to other datasets probably will be bottlenecked by training a good text-to-code converter and the complexity/ambiguity of the physics scene.
  -  In UTOPIA it is easy to obtain text code pairs for training the text-to-code converter since the dataset itself is generated by simulation. However, for other datasets (like PiQA or MMLU Physics) how can sufficient training data be obtained?
  - There are also complex physics scenes (e.g. liquid and deformable bottles etc in PiQA) that could be hard or lengthy to fully specify to a simulator. And the resulting physics metric to read out could also be more specific than only a binary comparison.
- The proposed method of injecting a physics simulator is conceptually very similar to the injection of calculator used in GSM8K dataset (Training Verifiers to Solve Math Word Problems). The specific idea of using physics simulation to help solve textual physics QA is also already explored by previous works (e.g. Grounding Physical Concepts of Objects and Events Through Dynamic Visual Reasoning; the physics here is simulated via a trained neural network to answer the more complex counterfactual questions with video input). These related previous works should also be cited and acknowledged.


**Summary Of The Paper:**

This paper aims at understanding and improving the ability of current LMs on reasoning over some basic laws of physics. A new text-based automatically generated physics QA dataset, UTOPIA, is proposed, and the authors showed that current LMs struggle on it. The authors then propose Mind's Eye to improve the physics reasoning ability by incorporating physics simulation results in the prompting. Specifically, a separate LM is trained to generate a simulation config from the original question, and the simulation result is appended to the prompt as a hint. With this method, LM zero-shot and few-shot performances are improved significantly across all scales reported and the improvement scales as the model size.

**Summary Of The Review:**

Overall, this paper presents a proof-of-concept way to combine external physics knowledge and the general reasoning ability of LM in a very simple setting. The demonstrated performance gain across scales is impressive and reveals important insights. However, the generalizability of the specific proposed method to more complex settings is unclear.

---

> ### Author Response · Authors · 2022-11-19
> **Response to Reviewer 46te**
>
> Thanks for reading our paper! We feel encouraged that you found our work clearly written and of high quality. We answer your questions as below:
>
> **Weakness 1: Domain Generalization.**
> Thanks for your question! Yes we agree that currently it might be difficult to obtain many more text-code pairs as training data for other domains, but we believe in the future it is solvable as we have already seen some progress in large-scale program synthesis models, such as Codex [1] from OpenAI, InCoder [2] from Meta, and CodeGen [3] from Salesforce. In preliminary experiments, we have tested using CodeEx to generate code by providing few-shot text-to-code demonstrations in the prompt, but we find the performance heavily depends on the choice of demonstrations and is not as good as training a specific text-code model for our task. The reason could be that the XML code used by MuJoCo is not very common in the pre-training data of these code models, which limits their in-context learning ability in our case. On the other hand, specifying the parameters of a simulation is usually much easier than writing a complete algorithm which program synthesis often needs to do.
>
> If Mind's Eye inspires the community and people try to render many text-form scenes by a physics engine either for grounded reasoning (e.g., using simulation to solve questions in PIQA and MMLU-Physics), robotics (where simulations and their natural descriptions are commonplace) or for entertainment (e.g., game industry, where people develop games to describe virtual worlds), then it is possible we can accumulate a large amount of text-to-code pairs from model libraries such as the MuJoCo Menagerie. The code in that case may not be MJCF files for MuJoCo, but instead C# files for Unity or Unreal (two successful game engines). In this work, we choose MuJoCo because its programs are succinct and good enough for physics reasoning. Higher-end game engines are capable of simulating more realistic scenes (with light chasing or particle effects). We leave that for future work when a detailed simulation is necessary.
>
>
> **Weakness 2: Complex Physics Scenes.**
> Thanks for your insightful comments! We agree there exist questions that are hard to be described by simulation, and a metrics readout may not be adequate enough for more complicated reasoning. These are actually limited by the chosen physics engine MuJoCo, since it still does not support fluid simulation, and it only supports raw sensory outputs without any summarization. We have added discussions of these problems in the newly added "Limitation" section (section 6).
>
> One immediate solution would be switching to a more advanced physics/game engine, or to add extra modalities: For example, the model not only relies on the text-from metrics output for accurate sensing, but also reads the video of simulation (with multi-modality LMs). And we believe multi-modality benchmarks might be necessary to overcome the ambiguity in text-only benchmarks (e.g., how much deformation on the bottles?). Ultimately, text-to-image models may even be a better alternative to text-to-code models for specifying the parameters of a simulation such as a relationship and relative positions of objects. As we have mentioned in the paper, the proposed Utopia will also come with simulation videos, which might support the multi-modality version of Mind's Eye. We leave that for future studies.
>
> **Weakness 3: More Related Work.**
> Thanks for the suggestion! In the revised "Related Work" section, we have included the citations of the papers you mentioned and many more works that learn the physics rules with neural models.
>
> Thanks again for your great suggestions and valuable feedback! Please tell us what else we can do to further improve our work!
>
> [1] [Evaluating Large Language Models Trained on Code](https://arxiv.org/pdf/2107.03374.pdf)
>
> [2] [InCoder: A Generative Model for Code Infilling and Synthesis](https://arxiv.org/pdf/2204.05999.pdf)
>
> [3] [CodeGen: An Open Large Language Model for Code with Multi-Turn Program Synthesis](https://arxiv.org/pdf/2203.13474.pdf)

---

### Official Review · Reviewer_TaNC · 2022-10-24

**Confidence:** 5
**Correctness:** 4
**Technical Novelty And Significance:** 4
**Empirical Novelty And Significance:** 4
**Recommendation:** 8

**Clarity, Quality, Novelty And Reproducibility:**

*Quality:*

This paper has high quality. See above strengths.

*Novelty:*

This paper is pretty novel

*Reproducibility:*

This paper doesn’t provide source code / interaction website.


**Strength And Weaknesses:**

*Strength:*

The idea proposed by this paper is pretty novel and effective;

This paper is well written and easy to understand;

This paper provides a new benchmark to test physic related reasoning tasks;

This paper has completed comparisons with current reasoning enhanced techniques, such as Chain of Thought, “Let’s think step by step” with zero-shot priming, Self-consistency decoding.

*Weaknesses:*

I hope this method could be open sourced, which will promote the development of the whole community.


**Summary Of The Paper:**

This paper proposes a new paradigm – adding a computational physics engine to aid language modeling on physical reasoning tasks.


**Summary Of The Review:**

This article greatly improves language models’ reasoning abilities on physical related tasks. This idea is pretty novel and this method consists of human physical reasoning steps: carry about simulation then write sentences about it.

---

> ### Author Response · Authors · 2022-11-19
> **Response to Reviewer TaNC**
>
> Thanks for reading our paper! We appreciate you find the method we proposed is pretty novel and the paper has high quality!
>
> About open-sourcing, we have included the dataset with our submission and code is still waiting for internal approval. And we thank you for your suggestion about an interaction website—yes we plan to extend this work and an interaction website is actually under development!
>
> Thanks again for your reviews!

---

### Official Review · Reviewer_pY5b · 2022-10-26

**Confidence:** 5
**Correctness:** 4
**Technical Novelty And Significance:** 3
**Empirical Novelty And Significance:** 3
**Recommendation:** 6

**Clarity, Quality, Novelty And Reproducibility:**

This paper is clearly written and seems fairly straightforward to reproduce assuming the authors release the dataset.


**Strength And Weaknesses:**

Strengths

1. This paper is well-motivated and attempts a simple and succinct approach towards adding in additional world information into a language model for physical reasoning tasks.
2. They create and release a dataset that focuses on physical reasoning tasks and use this to evaluate the models to see how they perform with augmented physical information
3. They illustrate and discuss how this is an attempt at a method that is both scalable and “grounded”---although it is worth noting that the inputs/outputs are still only ever in text format and the base LM has still never experienced input apart from text; however the additional context given by the physics simulator can serve as a proxy for non-textual information.


Questions / Weaknesses

1. Are there any insights on which reasoning abilities might be easier to gain information from from text (e.g., looking at the categories of motion, friction, free-fall etc.). This could be verified via co-occurrence statistics e.g., looking at which related words occur more in the training datasets, as well as which related answers might occur more in correspondence with the question in the prompt.
2. It might be good to have a memorisation control to test whether or not the answers generated by the model might have simply existed somewhere in it's training data (which means it's not actually using the outputs from the physics simulator to reason/generate the correct response).
3. Can the authors report error bars and variance in all the accuracy results?
4. Some missing citations in grounded reasoning for text-only LMs: Li et al., 2021; Patel and Pavlick, 2020.



**Summary Of The Paper:**

This paper attempts to create a pipeline that combines inputs of a language model with a computation physics engine to simulate possible outcomes, attempting to provide external world information into a text-only language model to assess whether this allows them to reason better.  They create a range of such tasks (as a benchmark dataset) and evaluate the improvement in performance across a range of model sizes. They see that this approach of giving in extra information from a physics engine as part of a language models context allows better performance, and additionally allows smaller language models to achieve performance at-par with the larger text-only ones.


**Summary Of The Review:**

Interesting evaluation paper and release of dataset.

---

> ### Author Response · Authors · 2022-11-19
> **Response to Reviewer pY5b**
>
> Thanks for reading our paper! We are glad that you find our work well-motivated and that the proposed method has the merit of being succinct and effective. We answer your questions as below:
>
>
> **Question 1: Which reasoning ability is easier to gain from text?**
>
> Thanks for your question! The reasoning abilities only gained from text are shown as the “vanilla” LMs in Table 3: For example, vanilla GPT-3 175B/1.3B performs differently in different categories in zero-shot/few-shot. In general, GPT-3 struggles with “free fall” questions as both 175B and 1.3B have the lowest zero-shot accuracy in the “free fall” category (i.e., 22.2% and 6.7%). We also find GPT-3 performs the best in the “projection” category among others. In the newly added Table A1 in the appendix we have also reported category-wise performance of Google’s Pathways Language Model (PaLM), and we see similar differences in different categories (e.g., the model performs the worst on “free fall” questions but pretty well on “projection” and “incline”.).
>
> Following your suggestions, we run a co-occurrence analysis on the Utopia benchmarks and the LM pre-training dataset. We ran the analysis on the dataset used to train PaLM. Since the dataset is quite huge and querying the whole dataset would be prohibitively time-consuming given the limited time window for rebuttal, we sample one million documents as a representative subset for a reasonable estimation. We first extract some “feature tokens” from each category of Utopia with TF-IDF (e.g., “free fall” as one of the feature tokens for free fall questions), and query these words in the training dataset. We show the number of/the ratio of feature tokens appearing in the training dataset as below:
>
> |          | Motion | Friction | Freefall | Projection | Collision | Incline |
> |----------|:------:|:--------:|:--------:|:----------:|:---------:|:-------:|
> | # Tokens |  6.3k  |   370k   |   6.8k   |    1944k   |    450k   |  9780k  |
> | % Tokens |   ≈0%  |  ≈0.69%  |    ≈0%   |   ≈0.36%   |   ≈0.08%  |  ≈1.83% |
>
> We find some correlation between the frequency of “feature words” and vanilla PaLM performance: For example, “projection” and “incline” are indeed mentioned rather frequently, and “free fall” feature words are relatively rare in the training dataset. Such correlations between term frequency and reasoning ability have also been reported in [4] recently.
>
> Note that such an exact matching query cannot handle high-level paraphrases so it can only serve as an estimation of how much text-form knowledge of a certain category is already perpetuated in the LM. In a word, without taking knowledge from simulation, the reasoning ability of LMs seems to be correlated with how much the LMs are familiar with a certain category.
>
>
> **Question 2: Memorization control to see whether the LM is using knowledge from simulation.**
>
> Thanks for your question! In Table 3, we have shown that before (vanilla GPT-3 175B, first row) and after using Mind’s Eye (Ours: Mind’s Eye with GPT-3 175B, the row with a star), the accuracy on different scenes categories achieve different gains. Vanilla GPT-3 only takes text-form questions as the input, while Mind’s Eye will inject additional rationales from simulations after the text-form questions. To see whether the LMs are actually using the knowledge from the simulation, we can simply compare the performance before and after the LM is augmented by Mind’s Eye. These results have been shown in the last row of Table 3 (i.e., “Grounding Gain”). From the results, we see significant gains across all categories (on average 27.9% and 46% improvement), indicating the LM indeed leverages the knowledge from simulation.
>
> **Question 3: Error bars and variance.**
>
> Thanks for your great suggestion! We have included the error bars in the revised Figure 1, and added the variance to all the accuracy results in the new Table A1 in the appendix (given the limited space we have to omit the average columns).
>
> **Question 4: More related work.**
>
> Thanks for your suggestion on related work! These are definitely related works and in the heavily revised "Related Work" section, we now have included the citations you mentioned [1, 2] in the “Grounded Reasoning” subgraph (Please correct us if we cite the wrong papers since you did not specify which one is Li et al. 2021), and we have also included many more works we think will be of readers’ interests.
>
> Finally, we appreciate the great suggestions and valuable feedback. We hope our revised version could solve your concerns. Thanks again for your thoughtful reviews!
>
> [1] [Implicit representations of meaning in neural language models.](https://aclanthology.org/2021.acl-long.143.pdf) Li et al. 2021
>
> [2] [Mapping Language Models to Grounded Conceptual Spaces.](https://openreview.net/pdf?id=gJcEM8sxHK) Patel and Pavlick, 2020

---

### Author Response · Authors · 2022-11-19
**General Response to All Reviewers**

We thank all the reviewers for their valuable comments and constructive feedback! We are glad that reviewers find strengths in our paper’s novelty (**R2**, **R3**), quality (**R2**, **R3**), and contribution to the community (**R1**). We have revised the paper according to the suggestions (highlighted in bronze in the paper). We summarize the highlights from the revision below and address each reviewer’s feedback separately as well.

- **More related work (R1, R3)**. We have heavily revised the related work section, by including other grounded reasoning works on LMs (R1), and more studies that try to teach models about physics in other ways (R3). We have also added more related works we think might be of interest to the readers.
- **Presentation improvement (R1)**. As suggested by R1, we now have included the error bars in the revised Figure 2 and variance in the newly added Table A1 in the appendix. We have also polished some expressions to further improve the presentation of our work.
- **Discussion on limitation (R3)**. To summarize our response to R3 on the limitation of our work, we have added a dedicated section (Section 6) to discuss some current limitations of Mind’s Eye.

Finally, we thank all the reviewers for their thoughtful comments and great suggestions! Please let us know if you have any further questions. We are more than happy to help!

---

### Decision · Program_Chairs · 2023-01-20

**Decision:**

Accept: poster

**Justification For Why Not Higher Score:**

Reviewer 46te pointed two limitations of this work which are valid concerns.

**Justification For Why Not Lower Score:**

There are overall positive reviews without many major issues. The author responses have properly addressed many weaknesses raised in the reviews.

**Metareview: Summary, Strengths And Weaknesses:**

There are overall positive reviews without many major issues. The author responses have properly addressed many weaknesses raised in the reviews.

Reviewer 46te pointed two limitations of this work which are valid concerns. However, the paper is still good as the first (successful) attempt in this direction.

**Note From Pc:**

if the above contains the word "oral" or "spotlight" please see: "oral" presentation means -> notable-top-5% and "spotlight" means -> notable-top-25%. As stated in our emails, we are disassociating presentation type from AC recommendations